# Hunting Blemishes: Language-guided High-fidelity Face Retouching Transformer with Limited Paired Data

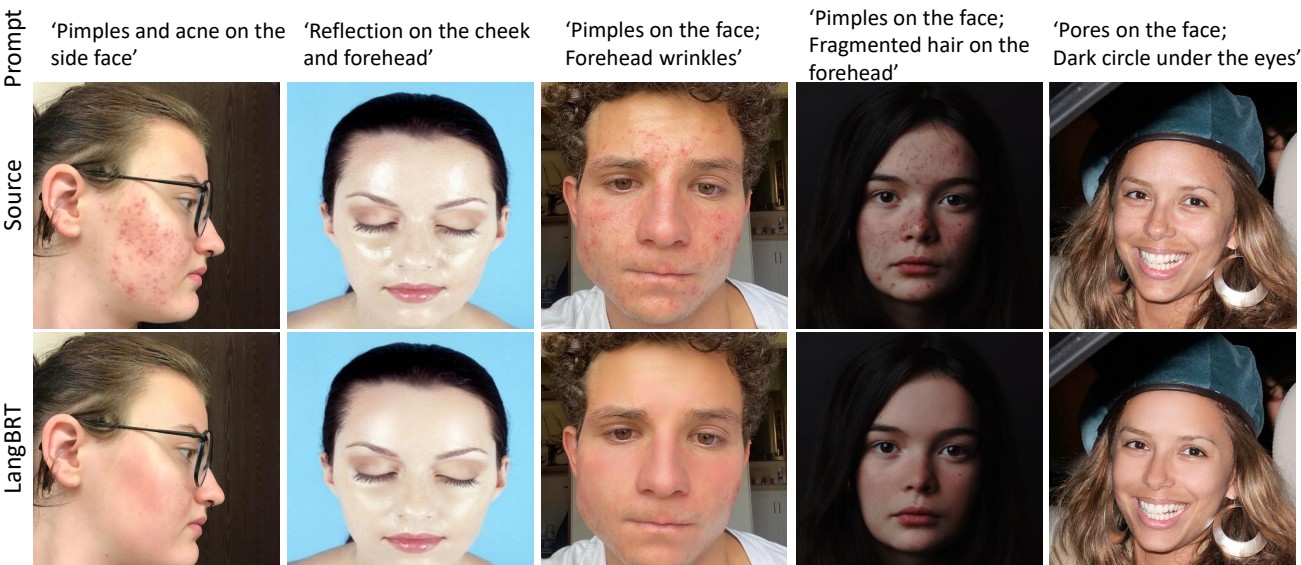

**Figure 1: Examples to illustrate the effectiveness of LangBRT in facial blemish removal. LangBRT is able to handle multiple types of blemishes in a variety of scenarios, while at the same time preserving non-blemish content as much as possible.**

## ABSTRACT

The prevalence of multimedia applications has led to increased concerns and demand for auto face retouching. Face retouching aims to enhance portrait quality by removing blemishes. However, the existing auto-retouching methods rely heavily on a large amount of paired training samples, and perform less satisfactorily when handling complex and unusual blemishes. To address this issue, we propose a Language-guided Blemish Removal Transformer for automatically retouching face images, while at the same time reducing the dependency of the model on paired training data. Our model is referred to as LangBRT, which leverages vision-language pre-training for precise facial blemish removal. Specifically, we design a text-prompted blemish detection module that indicates the regions to be edited. The priors not only enable the transformer network to handle specific blemishes in certain areas, but also reduce the reliance on retouching training data. Further, we adopt a target-aware cross attention mechanism, such that the blemish-like regions are edited accurately while at the same time maintaining the normal skin regions unchanged. Finally, we adopt a regularization approach

to encourage the semantic consistency between the synthesized image and the text description of the desired retouching outcome. Extensive experiments are performed to demonstrate the superior performance of LangBRT over competing auto-retouching methods in terms of dependency on training data, blemish detection accuracy and synthesis quality.

## CCS CONCEPTS

• **Computing methodologies → Computer vision tasks**.

## KEYWORDS

face retouching, transformer, vision-language pre-training, blemish detection

## 1 INTRODUCTION

The rapid development of social media leads to the fast-growing demand for automatic face retouching in various scenarios, including portrait photos, film and television productions, and so on. The primary objective of face retouching is to achieve natural-looking and realistic results, which maintains crucial characteristics while at the same time eliminating blemishes such as dark circles, acne scars and wrinkles [31, 46]. However, this is still a challenging task due to variations in lighting conditions, skin tones, and the complex nature of blemishes themselves.

Different from generic face enhancement tasks, there are typically a small percentage of image pixels that need to be edited in

face retouching, and the existing methods perform less satisfactorily due to lack of effective distinction between blemishes and normal skin. On the other hand, face retouching methods are built upon the observation that normal skin exhibits local smoothness. Many attempts have been made to design effective smoothing filters [1] to remove blemishes by leveraging the contextual information surrounding them. To handle diverse blemishes, deep convolutional neural networks are applied to learn the mapping from blemishes to normal skin [20, 30, 31, 44]. Further, generative models are trained to synthesize clean face images, conditioned on the ones with blemishes [14, 46]. These methods are trained on specific paired training data, thus limiting their generalization performance across difference domains where the appearance of blemishes and skin features, such as blemishes on different skin can vary significantly. Considering that the large-scale vision-language pre-training, such as CLIP [28], has strong capability of zero/few-shot object recognition, We perform an effective attempt for language-guided face retouching, and the resulting model can well generalize to diverse types of blemishes.

More specifically, we propose a Language-guided Blemish Removal Transformer (LangBRT) to facilitate face retouching. The key idea behind LangBRT is to perform textual prompt-conditional blemish detection and thus spatially regularize the cross-attention computation in transformer blocks to remove blemishes in a feature space, consequently generating a realistically retouched image corresponding to the prompt, as shown in Figure 1. To achieve this, we adopt the Contrastive Language-Image Pre-training model (CLIP) [28] to associate natural language with image content, and incorporate a Text-prompted Blemish Detection module (TBD), since a textual description can be used to effectively express rich visual concepts. TBD learns to perform pixel-wise recognition from the encoder features of an input face image. The prior knowledge encapsulated in CLIP enables TBD to distinguish blemishes from normal skin. On the other hand, We find that the prior is also useful for reducing the reliance of our model on paired training data. By injecting the resulting blemish feature maps as side information into the transformer, we can perform target-aware cross-attention computation, which aims to edit the blemish-like regions. We further impose the semantic consistency regularization on the synthesized images, given the textual description of the desired retouching outcome. Extensive experiments on both public and in-the-wild data are performed to verify the effectiveness of the design elements as well as the superior performance over state-of-the-art face retouching methods. In summary, the main contributions of this work are as follows:

- Different from the existing face retouching methods adopting generic image-to-image translation frameworks, the proposed LangBRT has a language-guided transformer architecture with target-aware cross-attention computation.
- Blemish detection is conditioned on the textual descriptions, which enable a wider range of blemish types to be effectively handled. Another benefit is to effectively reduce the dependency of LangBRT on paired training data.
- By injecting the blemish features into transformer blocks, the main feature transformations are limited in the blemish-like regions, which leads to precise retouching results.

## 2 RELATED WORK

### 2.1 Image-to-image Translation

Image-to-image translation can be viewed as a special case of conditional image generation, and there have been many attempts to facilitate this task. Given paired training data, a typical strategy is to train a convolutional neural network by minimizing a variety of regularization functions between the synthesized images and the ground truth [6, 16, 18, 49, 51]. In [6], a pixel-wise consistency loss to the ground truth was used for training an image translation network. To ensure semantic correctness, Johnson et al. [16] employed a pre-trained VGG-19 network [34] to measure the perceptual consistency, which leads to a better alignment with human perception in terms of image semantics. Similarly, Zhang et al. [49] proposed the Learnt Perceptual Image Patch Similarity (LPIPS) measure, which had been widely used in various image-to-image translation tasks. However, in many real-world scenarios, it is expensive and infeasible to collect a large amount of paired training data. Unsupervised image translation methods, like CycleGAN [51] and DiscoGAN [18], achieved impressive generation performance by minimizing the reconstruction loss of two-way mapping. By combining GAN [13] and VAE [19], UNIT [23] learnt to disentangle style and content in feature spaces, such that the images from different domains can be transferred by exchanging the style features.

To control the synthesis content, an additional attribute classifier was incorporated to guide the generation process by measuring the semantics encapsulated in the synthesized images [12]. StarGAN [7, 8] learnt a generation network to realize efficient cross-domain transformation, conditioned on domain label. Another effective way is to inject constraint information into the generation network, such as edges [15], sketches [2] and label maps [43]. Since the latent space of a well-trained StyleGAN [17] has a semantically meaningful organization, image-to-image translation can be performed by projecting a source image back into the latent space and learning a task-specific latent transformation. InterfaceGAN [32] was proposed to discover global latent directions associated with a number of pre-set attributes. To handle unlabeled data, Shen et al. [33] performed factorization on the weights of the generation network and found a set of latent directions associated with well-defined attributes.

### 2.2 Vision Transformer

Motivated by the great success of the transformer architectures in natural language processing [39], researchers have explored diverse applications of transformer in the computer vision area [10, 24, 38, 45]. The important characteristics of transformer lie in its attention mechanisms, which enable effective modeling of inherent relationships within sequences. In particular, ViT [10] extended the transformer's success to visual tasks. To further enhance the ViT's capabilities in handling high resolution images, Wu et al. [45] proposed a convolutional vision transformer, which incorporated convolutional layers into transformer blocks. On the other hand, SwinTransformer [24] adopted the shifted windowing scheme, which limited attention calculation to non-overlapping windows while allowing cross-window connection. Transformer architectures were also successfully applied to object detection,

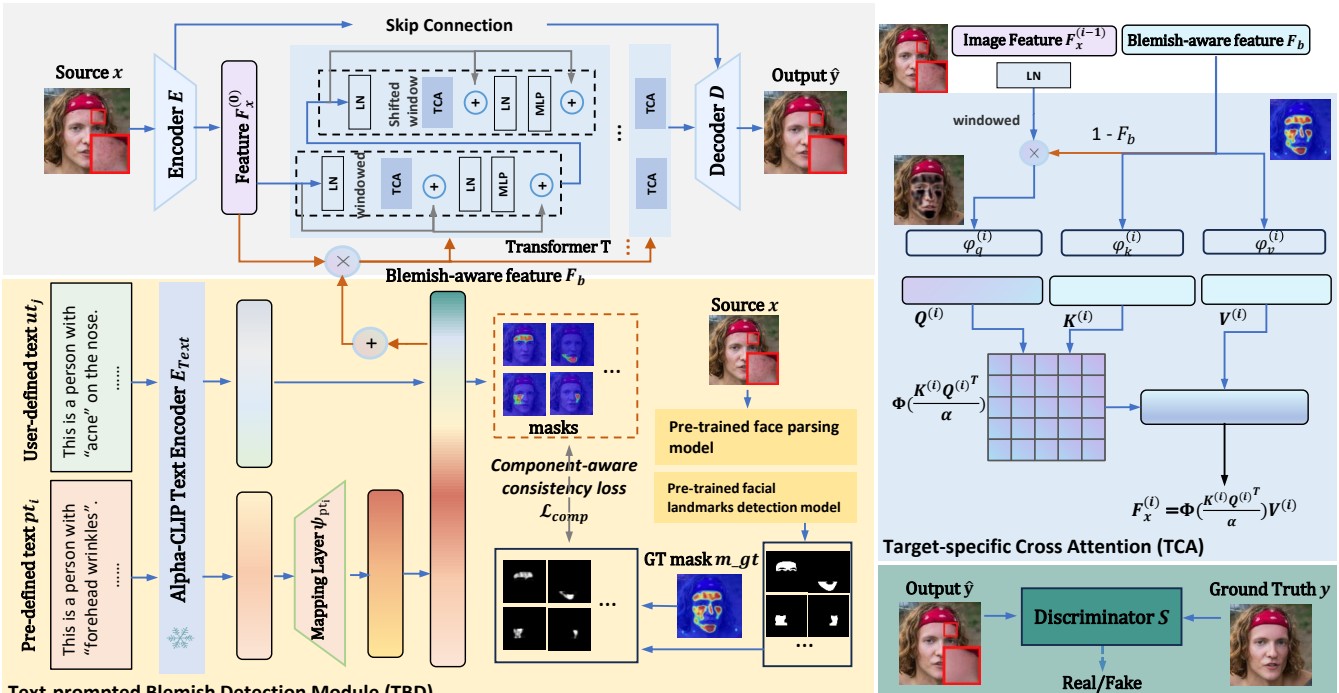

Figure 2: An Overview of the proposed LangBRT. An image encoder $E$ and the pre-trained Alpha-CLIP [35] text encoder $E_{Text}$ are used to extract features from the input image and the textual description of blemishes, respectively. TBD aims to detect the specific blemishes associated with the textual prompts. The detection maps are integrated and then fed into a latent transformer $T$, in which we perform TCA in each block to progressively transform the features associated with the blemishes. Finally, the decoder $D$ is used to generate a clean face image from the transformed features.

such as DETR and variants [4, 52]. Inspired by DETR, Wang et al. [41] proposed an end-to-end segmentation transformer to directly predict masks with class labels. Contrastive Language-Image Pre-Training (CLIP) [28] was designed to understand and associate images and textual descriptions, and had gained significant attention due to its versatility and effectiveness. The CLIP's capability of performing cross-modal understanding together with transformers had been widely used in various applications, such as image generation [29], image classification[5], retrieval[25], semantic segmentation [21], video caption [36], video action recognition [42] and object localization [9]. To apply CLIP on downstream tasks, Gao et al. proposed CLIP-Adapter[11] to conduct fine-tuning with feature adapters on either visual or language branch. Furthermore, Sun et al. [35] proposed Alpha-CLIP to enhance CLIP with an auxiliary alpha channel to suggest attentive regions, which enables Alpha-CLIP to focus more on the regions of interest.

The objective of face retouching is to enhance the appearance of input images while preserving the key facial characteristics. The traditional methods, like nonlinear digital filtering [1], applied a uniform operation to address different types of flaws. In [3], Batool et al. detected facial wrinkles and imperfections using Gabor filters. In addition, Velusamy et al. [40] proposed a wavelet band manipulation method to restore the underlying skin texture. However, these methods lacked adaptive retouching capabilities. To address this issue, Lipowezky et al. [22] performed freckle detection and retouching separately. Based on the concept of facial attractiveness,

the face retouching process could be guided by an aesthetic enhancement model [37]. Recently, AutoRetouch [31] was an effective attempt to perform end-to-end face retouching. In addition, Zamir et al. [48] proposed a multi-stage approach to progressively restore spatial details and high-level contextualized information. These methods primarily focus on global retouching while neglecting the importance of the local region. Instead, ABPN [20] performed fast local retouching on high-resolution photos through an adaptive blend pyramid network. To guide precise blemish removal while preserving the semantic information of an input image, Hong et al. incorporated a pre-trained face parsing model in HQRetouch [14]. In contrast, BPFRe[46] was a multi-stage approach for face retouching, which divided the retouching process into blemish detection, retouching and refinement phases, and adopted different strategies to utilize unpaired training data to regularize each stage.

The main differences between our proposed LangBRT and the above existing face retouching methods are summarized as follows: (1) LangBRT facilitates face retouching by utilizing textual descriptions of blemishes, and it is the first attempt to leverage vision-language pre-training for the task. LangBRT is able to address diverse blemish types, and allows user-defined retouching. This has not been considered by the above methods. (2) Different from the existing methods [14, 20, 46] which directly suppressed features of blemish-like regions, LangBRT limited main feature transformations in blemish-like regions via target-aware cross attention, which ensures precise blemish removal.

# 3 METHODOLOGY

## 3.1 Overview

It is promising to integrate the description of blemishes together with desired retouching outcome into our image editing process. As shown in Figure 2, the proposed framework mainly consists of five components, including an encoder $E$ extracting the features from a source image, a latent transformer $T$ performing feature transformation, a decoder $D$ generating a clean face image, a discriminator $S$ distinguishing manually retouched images from the synthesized ones, and a Text-prompted Blemish Detection module (TBD). Given the blemish descriptions involving dark circle, acne, wrinkle and so on in specific area, TBD leverages a pre-trained vision-language model to obtain the textual prompts together with an image encoder to extract features from a source image, and produces the corresponding maps to indicate the blemishes associated with the descriptions, respectively. Further, we inject the blemish information into the transformer blocks in $T$ via Target-specific Cross-Attention mechanism (TCA) to limit the main feature transformations in the blemish-like regions, and the transformer blocks are guided to progressively restore clean skin in the regions.

## 3.2 Text-prompted Blemish Detection

In LangBRT, prompt is defined as the combination of blemish type and its corresponding location. To streamline model training, we delineate two categories of prompts: (1) The first category is pre-defined prompt, such as 'dark circles under the eyes', 'forehead wrinkles', 'acne on the right cheek'. Pre-defined prompts encapsulate common blemishes and cater to the general needs of face retouching. To enhance the localization accuracy of these blemishes across diverse inputs, we incorporate dedicated mapping layers for each pre-defined prompt. (2) The second category is user-defined prompt, which is tailored to accommodate users' personalized preferences for precision retouching. Given the variability in refinement requisites among users, these prompts provide users with added control over the retouching process.

Let $[\boldsymbol{pt}_1, \boldsymbol{pt}_2, \ldots, \boldsymbol{pt}_P]$ denote a set of pre-defined prompts, and $[\boldsymbol{ut}_1, \boldsymbol{ut}_2, \ldots, \boldsymbol{ut}_U]$ denote a set of user-defined prompts. We adopt the CLIP text encoder $E_{Text}$ of the pre-trained Alpha-CLIP[35] to generate corresponding embeddings $[F_{\boldsymbol{pt}_1}, F_{\boldsymbol{pt}_2}, \ldots, F_{\boldsymbol{pt}_P}]$ and $[F_{\boldsymbol{ut}_1}, F_{\boldsymbol{ut}_2}, \ldots, F_{\boldsymbol{ut}_U}]$. Let the mapping layer $\psi_{\boldsymbol{pt}_i}$ to learn the precise text features corresponding to the pre-defined prompt. Let $\boldsymbol{x}$ denote a source image, which is passed through the image encoder $E$ to extract the feature $F_{\boldsymbol{x}}^{(0)}$. The manually retouched image denoted as $\boldsymbol{y}$ serves as the ground truth. To detect the blemishes associated with the specified prompts, $E$ is encouraged to capture the blemish information from the image, and the detection map $\boldsymbol{m}$ is derived by measuring the relevancy between the image features and each prompt as follows:

$$\boldsymbol{m} = \sum_{j=1}^{U} F_{\boldsymbol{x}}^{(0)} \odot \mathcal{E}(F_{\boldsymbol{ut}_j}) + \sum_{i=1}^{P} F_{\boldsymbol{x}}^{(0)} \odot \psi_{\boldsymbol{pt}_i}(\mathcal{E}(F_{\boldsymbol{pt}_i})), \quad (1)$$

where $\odot$ represents the dot product operation, $\mathcal{E}$ denotes the operation to expand the dimensionality of the text embedding to match that of the image feature, and $F_{\boldsymbol{x}} \odot \mathcal{E}(F_{\boldsymbol{ut}_j})$ refers to the blemish map associated with the $j$-th user-defined prompt $\boldsymbol{ut}_j$,

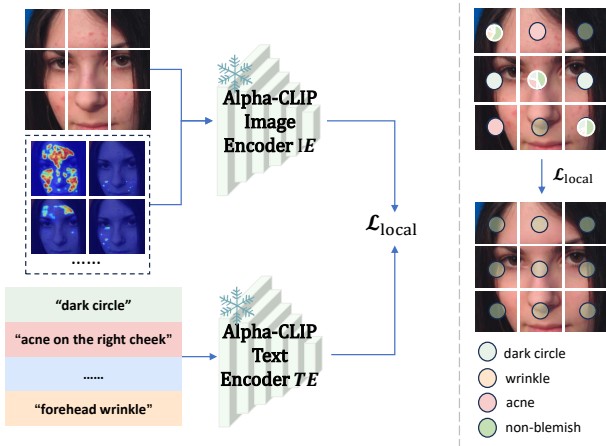

Figure 3: An example to illustrate the Alpha-CLIP-based semantic regularization. The training goal is to maximize the dissimilarity between the generated image and the textual prompts of blemishes in the Alpha-CLIP embedding space.

$F_{\boldsymbol{x}}^{(0)} \odot \psi_{\boldsymbol{pt}_i}(\mathcal{E}(F_{\boldsymbol{pt}_i}))$ refers to the blemish map associated with the $i$-th pre-defined prompt $\boldsymbol{pt}_i$, U and P represents the number of user-defined prompt and pre-defined prompt, respectively. We consider that the CLIP embedding space encapsulates rich knowledge on blemishes, and the prompt is an effective representation to retrieve the useful priors for our detection task. This design also helps to reduce the dependency on large amounts of manually retouched data.

## 3.3 Target-aware Cross Attention

We realize blemish removal by progressively replacing the blemishes with the synthesized content. Toward this end, we adopt a target-specific cross attention mechanism in each transformer block. Different from generic self-attention computation over all pixel positions, our mechanism limits the main feature transformations in blemish-like regions, while at the same time ensuring that the features in normal skin regions remain unchanged.

The blemish detection maps provide the spatial information on the regions to be edited. In each transformer block, the maps play two different roles in constructing the query and key-value vectors for attention computation. Specifically, the image features contain crucial visual details and patterns, and serve as the input of query mapping. The maps are used as the weight to suppress the features in the blemish-like regions. On the other hand, we directly learn key and value vectors from the maps, such that the blemish-like regions will be filled with the content synthesized from scratch. Since the features of blemishes are continuously discarded in the forward process, the learnt key and value vectors are associated with the features of normal skin to ensure clean face synthesis. Formally, we define the cross attention computation in the $i$-th transformer block as follows:

$$Q^{(i)} = \varphi_{\mathrm{q}}^{(i)}(F_{\boldsymbol{x}}^{(i-1)} \odot (1 - \boldsymbol{m})), \quad (2)$$

$$K^{(i)} = \varphi_{\mathrm{k}}^{(i)}(\boldsymbol{m}) \,, \, V^{(i)} = \varphi_{\mathrm{v}}^{(i)}(\boldsymbol{m}), \quad (3)$$

$$F_{\boldsymbol{x}}^{(i)} = \phi\left(\frac{K^{(i)} \cdot Q^{(i)\mathrm{T}}}{\alpha}\right) \cdot V^{(i)}, \quad (4)$$

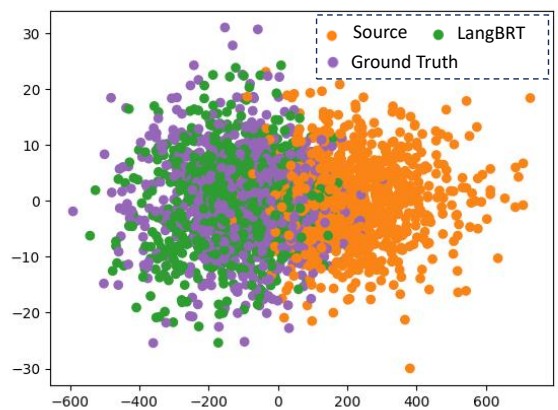

**Figure 4: Visualizing the distributions of source FFHQR images, ground-truth retouching images and images synthesized by LangBRT.**

where $F_x^{(i)}$ denotes the image features in the $i$-th transformer block for $i = 1, 2, \ldots, N$, $\{Q^{(i)}, K^{(i)}, V^{(i)}\}$ represent the query, key and value features, $\{\varphi_q^{(i)}, \varphi_k^{(i)}, \varphi_v^{(i)}\}$ are the corresponding linear mapping functions, $\phi$ is the sigmoid activation function, and $\alpha$ is a learnable scaling parameter to control the magnitude of the multiplication result. To reduce the computational complexity, we adopt the sliding window strategy of SwinTransformer [24]. The final output of the transformer is fed into the decoder to synthesize a retouched image denoted as $\hat{y}$.

### 3.4 Model Training

The training goal of the proposed LangBRT consists of two aspects: precise blemish detection and high-fidelity retouched image synthesis. Due to the lack of blemish annotations, we compute the difference map between each pair of raw image and manually retouched one, which is used as the ground truth of blemish detection. Conditioned on the textual prompts, our detection model produces a set of detection maps, and their combination is required to be consistent with the ground truth. The detection loss function can be defined as follows:

$$\mathcal{L}_{detc} = \mathbb{E}_x[||\boldsymbol{m} - \tau(|\boldsymbol{x} - \boldsymbol{y}|)||_1], \quad (5)$$

where $\tau$ is an activation function to normalize the difference map. Minimizing $\mathcal{L}_{detc}$ encourages the encoder to capture the information on blemishes. Furthermore, for the blemishes in the particular domain, we roughly segment the face based on existing face parsing model [50], as an auxiliary method for the model to refine the region-specific retouching. We obtained coarse segmentation of facial regions such as the forehead, left cheek, and periocular area. Subsequently, we applied component-aware consistency loss exclusively to these regions' blemishes, which is defined as follows:

$$\mathcal{L}_{comp} = \mathbb{E}_x\left[\sum_{m_{pt_i}}^{P}(||m_{pt_i}(\boldsymbol{m} - \tau(|\boldsymbol{x} - \boldsymbol{y}|))||_1)\right], \quad (6)$$

where $m_{pt_i}$ refers to the mask corresponding to pre-defined prompt $pt_i$, P is the number of pre-defined prompt.

To ensure high-fidelity image synthesis, we adopt an adversarial training approach to optimize the constituent networks. The synthesized image $\hat{y}$ is expected to be identified as a manually retouched one, and the discriminator aims to identify them as accurately as possible. We define the adversarial training loss functions as follows:

$$\mathcal{L}_{adv}^G = \mathbb{E}_x[log(1 - S(\hat{\boldsymbol{y}}))], \quad (7)$$

$$\mathcal{L}_{adv}^S = \mathbb{E}_y[log(\boldsymbol{y})] + \mathbb{E}_x[log(1 - S(\hat{\boldsymbol{y}}))], \quad (8)$$

where $S(\cdot)$ denotes the probability of an input image being retouched manually.

Considering that deceiving the discriminator cannot guarantee the retouching quality, we further measure the pixel-wise and perceptual consistency between the synthesized result $\hat{y}$ and the manually retouching image $y$, and the corresponding loss function is defined as follows:

$$\mathcal{L}_{cons} = \mathbb{E}_x[||\hat{\boldsymbol{y}} - \boldsymbol{y}||_1] + \beta\mathbb{E}_x\left[\sum_l ||\Phi_l(\boldsymbol{y}) - \Phi_l(\hat{\boldsymbol{y}})||_1\right], \quad (9)$$

where $|| * ||_p$ represents $\ell_p$ norm, $\Phi_l$ denotes the features associated with the $l$-th layer of a pre-trained VGG-19 [34] network, and $\beta$ is the weighting factor to balance the two types of consistency measurements.

In addition to leveraging the textual prompts for blemish detection, we can also use the textual descriptions of desired retouch outcomes to regularize the generation process by measuring the semantic similarity between the synthesized results, specific region and the descriptions in the Alpha-CLIP [35] embedding space. For simplicity, we still use the blemish descriptions and train the model by maximizing the prior-based dissimilarity to the synthesized results as follows:

$$\begin{aligned}\mathcal{L}_{local} = &\mathbb{E}_x\left[\sum_i^P \varrho_{\alpha CLIP}(\hat{\boldsymbol{y}}, m_{pt_i}, \boldsymbol{pt}_i)\right] \\ &+ \mathbb{E}_x\left[\sum_j^U \varrho_{\alpha CLIP}(\hat{\boldsymbol{y}}, m_{ut_j}, \boldsymbol{ut}_j)\right],\end{aligned} \quad (10)$$

where $\varrho_{\alpha CLIP}$ is pre-trained Alpha-CLIP. As shown in Figure 3, the prior-based dissimilarity loss function is useful for guiding the generation process.

By integrating the above training loss functions, the optimization formulation of LangBRT can be expressed as follows:

$$\begin{aligned}\min_{E,T,D} \quad &\mathcal{L}_{adv}^G + \mathcal{L}_{cons} - \gamma\mathcal{L}_{local} + \eta(\mathcal{L}_{detc} + \mathcal{L}_{comp}), \\ \max_S \quad &\mathcal{L}_{adv}^S,\end{aligned} \quad (11)$$

where $\gamma$ and $\eta$ are the weighting factors to achieve a trade-off among the regularization terms. Note that the constituent networks are jointly optimized from scratch. The training procedure is summarized in Appendix.

## 4 EXPERIMENTS

In this section, extensive experiments are performed to assess the retouching performance of the proposed LangBRT on both public and in-the-wild data. We first introduce the training and test data, implementation details, and evaluation protocol. Next, we compare LangBRT with state-of-the-art face image editing methods. Finally, we perform ablation study to verify the effectiveness of the main components.

Table 1: Quantitative comparison between LangBRT and competing methods on FFHQR. Boldface indicates the best results.

| Method | FFHQR-1% | | | FFHQR-5% | | | FFHQR-10% | | | FFHQR-20% | | | FFHQR-100% | | |
|---|---|---|---|---|---|---|---|---|---|---|---|---|---|---|---|
| | PSNR↑ | SSIM↑ | LPIPS↓ | PSNR↑ | SSIM↑ | LPIPS↓ | PSNR↑ | SSIM↑ | LPIPS↓ | PSNR↑ | SSIM↑ | LPIPS↓ | PSNR↑ | SSIM↑ | LPIPS↓ |
| Pix2PixHD[43] | 25.59 | 0.7711 | 0.1585 | 26.68 | 0.7963 | 0.1501 | 27.13 | 0.8008 | 0.1427 | 28.88 | 0.8526 | 0.1054 | 29.38 | 0.9181 | 0.0766 |
| GPEN[47] | 42.70 | 0.9872 | 0.0311 | 42.92 | 0.9884 | 0.0223 | 42.98 | 0.9895 | 0.0169 | 43.04 | 0.9901 | 0.0143 | 43.12 | 0.9903 | 0.0141 |
| SwinTransformer[24] | 41.92 | 0.9840 | 0.0353 | 42.10 | 0.9845 | 0.0309 | 42.29 | 0.9851 | 0.0235 | 42.53 | 0.9863 | 0.0199 | 43.19 | 0.9878 | 0.0130 |
| AutoRetouch[31] | 38.49 | 0.9728 | 0.0161 | 39.64 | 0.9780 | 0.0144 | 41.11 | 0.9791 | 0.0140 | 42.22 | 0.9801 | 0.0135 | 44.18 | 0.9804 | 0.0133 |
| MPRNet[48] | 42.12 | 0.9874 | 0.0311 | 42.57 | 0.9889 | 0.0242 | 43.29 | 0.9901 | 0.0144 | 43.52 | 0.9901 | 0.0137 | 44.35 | 0.9907 | 0.0129 |
| RestoreFormer[44] | 39.87 | 0.9791 | 0.0178 | 41.12 | 0.9802 | 0.0164 | 42.47 | 0.9879 | 0.0155 | 42.86 | 0.9900 | 0.0132 | 42.95 | 0.9904 | 0.0129 |
| ABPN[20] | 42.09 | 0.9862 | 0.0329 | 42.78 | 0.9887 | 0.0259 | 43.28 | 0.9895 | 0.0234 | 43.66 | 0.9903 | 0.0121 | 44.41 | 0.9918 | 0.0169 |
| BPFRe[46] | 43.19 | 0.9889 | 0.0129 | 44.22 | 0.9895 | 0.0125 | 44.50 | 0.9901 | 0.0110 | 45.01 | 0.9906 | 0.0109 | 45.29 | 0.9935 | 0.0092 |
| LangBRT | **44.51** | **0.9930** | **0.0113** | **45.07** | **0.9936** | **0.0101** | **45.30** | **0.9937** | **0.0096** | **45.41** | **0.9938** | **0.0092** | **45.72** | **0.9941** | **0.0086** |

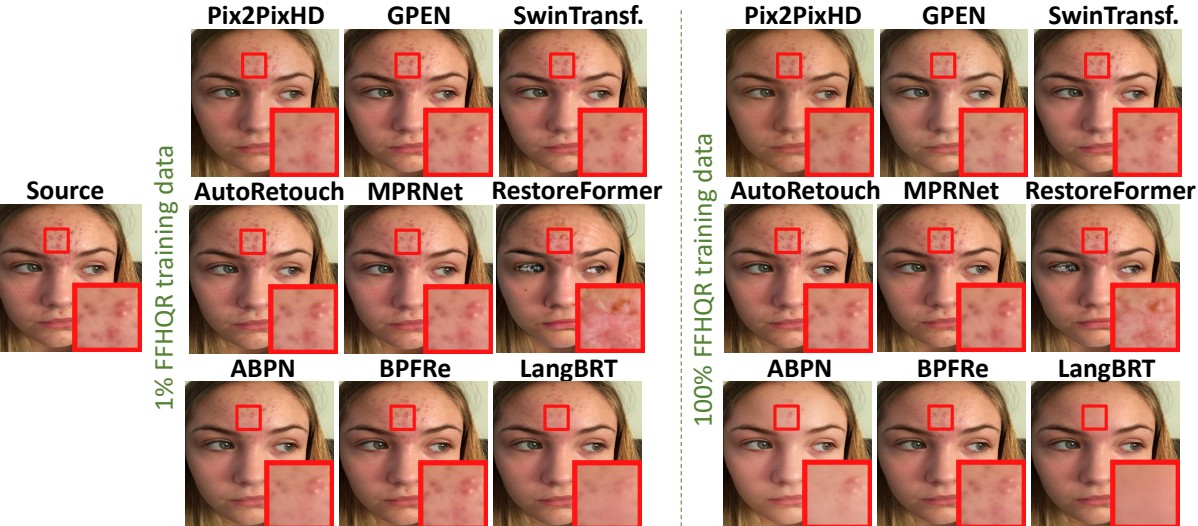

Figure 5: Visual comparison between LangBRT and competing methods on FR-wild dataset. (*Left*) The retouching results of the models optimized on 1% of the FFHQR training data. (*Right*) The retouching results of the models optimized on the whole FFHQR training data. LangBRT is able to achieve stable retouching performance.

## 4.1 Experimental settings

*4.1.1 Datasets.* The main experiments are performed on the FFHQR dataset [31], which is the first large-scale public dataset created through professional retouching techniques. It contains 70,000 pairs of "Before" and "After" retouched images with the resolution of 1024*1024, involving various facial characteristics such as age and race. The dataset is partitioned into a training, validation, and test set, containing 56,000, 7,000, and 7,000 images respectively. We follow the setting [31], in which both training and test images are resized to 512*512. Furthermore, the proposed method is also evaluated on the FaceRetouch-wild (FR-wild) dataset, which contains 700 in-the-wild face images with a large diversity of poses, races, and blemish types.

*4.1.2 Implementation Details.* In LangBRT, the latent transformer contains 7 blocks, and the configurations are the same as Swin-Transformer [24]. The architecture information of the transformer together with the other constituent networks are provided in Appendix. We implement LangBRT using PyTorch with a NVIDIA GeForce RTX 3090. There are a total of 25,000 training iterations with a batch size of 1. The number of user-defined prompt and pre-defined prompt, U and P in Eq.(1) is set to 17 and 3, respectively. The weighting factors: $\beta$ in Eq.(9) and $\{\gamma, \eta\}$ in Eq.(11) are set to 10, 1 and 1, respectively. Our model is optimized through Adam, and

the learning rate is initialized as 0.0002 and modified according to a cosine decay schedule.

## 4.2 Quantitative Comparison

We compare the proposed LangBRT with a number of representative face image editing methods, including Pix2PixHD [43], GPEN [47], SwinTransformer [24], MPRNet [48], RestoreFormer, [44], AutoRetouch [31], ABPN [20], and BPFRe [46]. Note that SwinTransformer serves as the base model of our LangBRT. Pix2PixHD is a typical image-to-image translation method. MPRNet and RestoreFormer (GPEN) are designed for (face) image restoration. AutoRetouch, ABPN and BPFRe focus on face retouching. We follow the settings of BPFRe to perform quantitative evaluation in terms of Peak Signal-to-Noise Ratio (PSNR), Structural Similarity Index Measure (SSIM), and Learnt Perceptual Image Patch Similarity (LPIPS).

We implement all the competing methods using the open source codes, and they are trained on the same data as our LangBRT for a fair comparison. In addition to using the full training data, we also randomly sample 1%, 5%, 10%, and 20% of the training data to evaluate the performance stability of the competing method in the situations of limited training data. The results are summarized in Table 1. We can observe that LangBRT consistently outperforms the competing methods in terms of all the metrics. When using

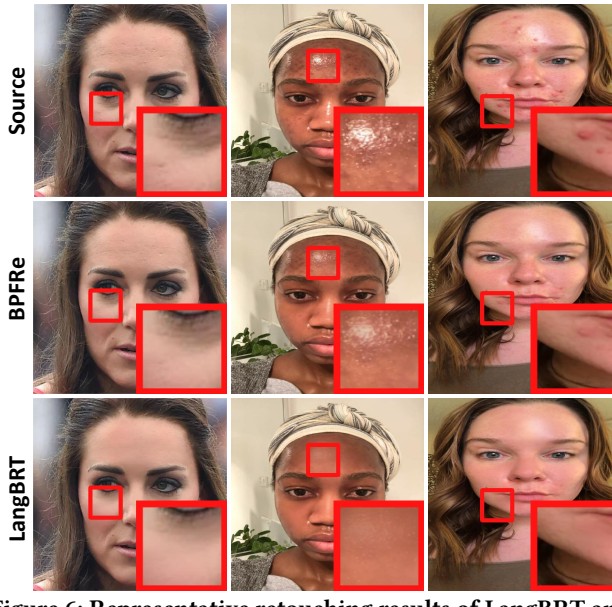

**Figure 6: Representative retouching results of LangBRT and BPFRe. LangBRT is capable of producing more satisfactory retouching results in removing different types of blemishes, compared to BPFRe.**

only 1% of the training data, the advantage of LangBRT becomes more significant. In particular, LangBRT achieves the PSNR score of 44.51 dB, which is higher than that of BPFRe by about 1.32 dB. The SSIM and LPIPS scores of LangBRT are better than the second best methods: BPFRe, by about 0.41 and 0.16 percentage points. In addition, we additionally adopted the structure of the Temporary Patch GAN model [27], trained a classifier for source and ground truth images of FFHQR dataset, and visualized the features of the intermediate layer using PCA [26], as shown in Figure 4. Notably, it can be seen that the refined results of LangBRT are similar to the ground truth, but there are significant differences from the source images, which also proves the effectiveness of LangBRT. We consider that our model benefits from the vision-language pre-training, and thus has a lower dependence on the training data than the competing methods. This is also confirmed by the representative retouching results shown in Figure 5.

## 4.3  Qualitative Comparison

According to the above quantitative comparison result, BPFRe performs better than the other competing methods. To highlight the LangBRT's capability of handling diverse blemishes, we further compare with the state-of-the-art face retouching method BPFRe in Figure 6. One can find that LangBRT is able to remove dark circles and acne, reduce reflections, smooth skin, while preserving the original tone. In contrast, BPFRe performs less satisfactorily, and the blemishes are only partially removed. This result suggests that LangBRT has better generalization performance in real-world scenarios. We further perform user study to assess the retouching performance of the methods in human perception. We randomly sample 30 face images from the in-the-wild data, and ask 50 volunteers to rank the synthesized results of LangBRT and the competing methods. All the models are trained on 1% of the training data in FFHQR.

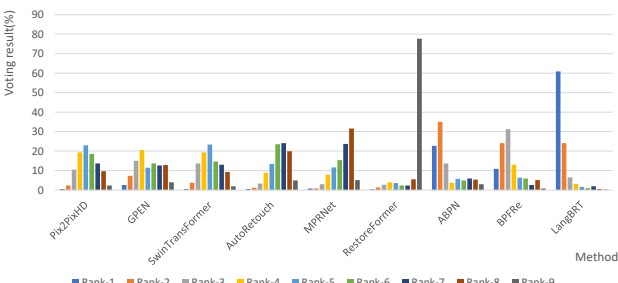

**Figure 7: The voting result (%) of user study on FR-wild.**

We provide a visual representation of the comparative preferences expressed by the participants in Figure 7. The result suggests that LangBRT receives the most votes as the best retouching method, and this is consistent with the results obtained from the previous experiments, further validating the superiority of LangBRT in our task.

## 4.4  Ablation study

To investigate the contribution of the main components in LangBRT, we build a number of variants using 1% of the training data and perform ablative experiments in this subsection. (1) " LangBRT w/o Prior": the vision-language pre-training is not used for blemish detection and semantic regularization. (2) " LangBRT w/o TBD": the text-prompted blemish detection module is disabled. (3) " LangBRT w/o TCA": the target-specific cross attention is replaced with the generic self-attention mechanism in each transformer block. (4) " LangBRT w/o $\mathcal{L}_{local}$": the loss function $\mathcal{L}_{local}$ is disabled. The retouching performance of the variants are summarized in Table 2 and Figures 8, 9 & 10. Table 2 shows that the removal of the vision-language pre-training results in a significant increase in LPIPS by over 9 times. We consider that the pre-trained model plays an important role in providing useful priors of blemishes in the limited data case. Without accurate blemish detection or target-specific cross attention, " LangBRT w/o TBD" or " LangBRT w/o TCA" cannot limit the main feature transformations in blemish-like regions, such that the information from non-blemish regions cannot be effectively utilized for filling the blemish-like regions, and the retouching performance thus becomes less satisfactory as shown in Figure 8. In addition, we confirm that $\mathcal{L}_{local}$ is useful for boosting the performance by 0.72dB in terms of PSNR (Table 2). We plot the PSNR curve of the model with and without $\mathcal{L}_{local}$ during the training propose, and find that the loss function consistently leads to higher PSNR values(Figure 9). Furthermore, we visualize the feature changes of representative images before and after processed by Transformer $T$, the results shown in Figure 10 suggest that $\mathcal{L}_{local}$ enhances the transformer's precision in refining defect areas.

## 4.5  Customized Prompts for Blemish Removal

LangBRT has the capability of detecting and removing the blemishes associated with the user-defined textual descriptions. To illustrate the effectiveness of the text-prompted blemish detection module, we visualize the detection maps involving different prompts in Figure 11. It can be observed that the module produces different detection maps when using the prompts: "dark circle", "forehead wrinkles" and "pimple on the right cheek". Furthermore, LangBRT

Source — LangBRT w/o Prior — LangBRT w/o TBD — LangBRT w/o TCA — LangBRT w/o $\mathcal{L}_{local}$ — LangBRT

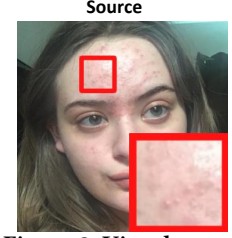 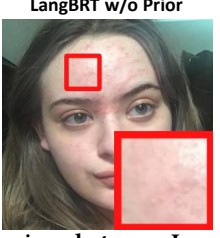 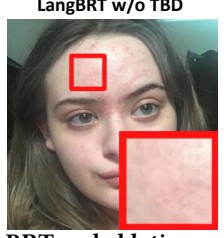 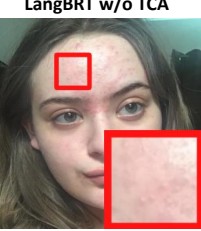 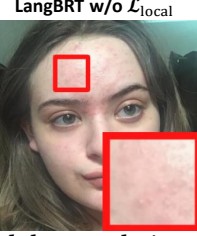 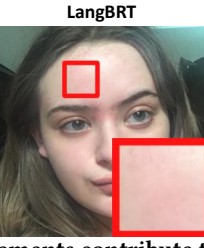

Figure 8: Visual comparison between LangBRT and ablative models. It can be observed that our design elements contribute to the final retouching performance of LangBRT.

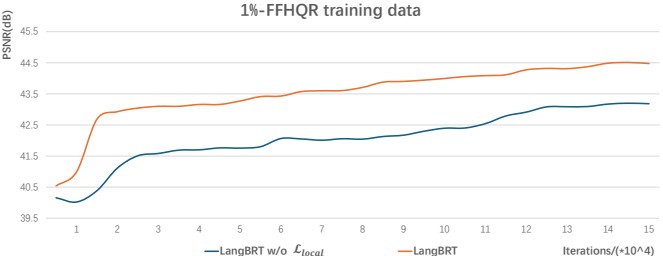

Figure 9: Illustrating that LangBRT is able to faster converge to a better solution than LangBRT w/o $\mathcal{L}_{local}$ on 1% training dataset.

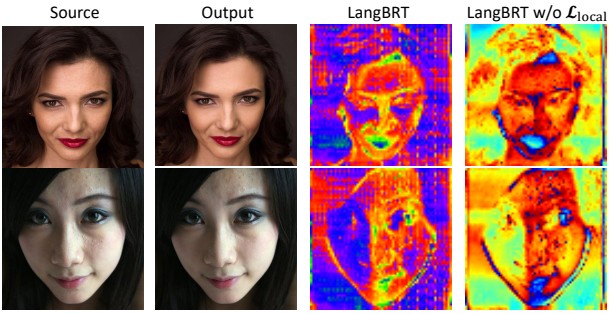

Figure 10: Visualization of the feature changes before and after processed by Transformer $T$ for the cases of with and without $\mathcal{L}_{local}$. Brighter regions indicate larger feature alterations, while darker regions indicate minor changes.

Table 2: Quantitative results of LangBRT and ablative models on FFHQR-1%.

| Method | PSNR↑ | SSIM↑ | LPIPS↓ |
|---|---|---|---|
| LangBRT w/o Prior | 42.61 | 0.9904 | 0.0985 |
| LangBRT w/o TBD | 42.80 | 0.9911 | 0.0129 |
| LangBRT w/o TCA | 43.64 | 0.9921 | 0.0128 |
| LangBRT w/o $\mathcal{L}_{local}$ | 43.19 | 0.9919 | 0.0122 |
| LangBRT | 44.51 | 0.9930 | 0.0113 |

is able to remove the blemishing accordingly. In Figure 12, we compare with BPFRe and human retouchers, and find that the detection maps of our model are more consistent with the manual results.

## 5 CONCLUSION

This paper presents a text-driven latent transformer for precise facial blemish detection and removal. To handle diverse blemish

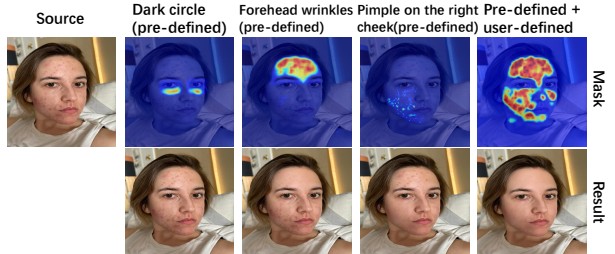

Figure 11: The representative results of blemish detection conditioned on different textual prompts.

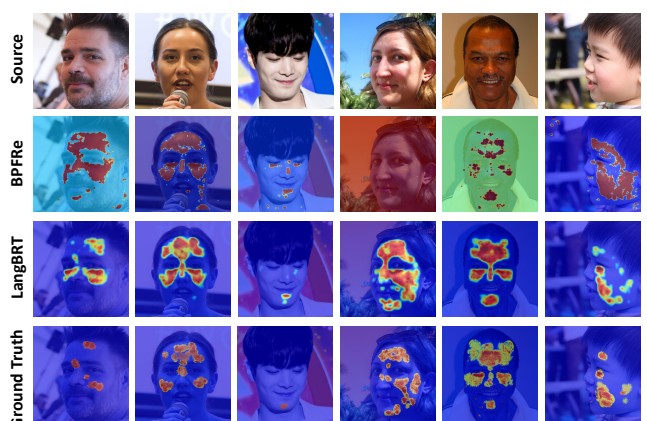

Figure 12: Visual comparison between LangBRT and BPFRe in blemish detection.

types, we incorporate the vision-language pre-training and leverage the prior knowledge encapsulated in the embedding space. By providing the textual prompts of blemishes, we design an effective detection module to measure the association between the encoder features and textual prompts, and produce the maps to highlight the spatial information of specific blemishes. This design not only improves the generalization performance on a wide range of blemishes, but also reduces the dependence of the model on the paired training data. To precisely remove blemishes while preserving non-blemish content, we further inject the blemish map into each transformer block to perform target-aware attention computation. In the forward process, the features of blemish-like regions are replaced with the synthesis content progressively. Extensive experiments demonstrate the superiority of the proposed method over the state-of-the-arts, especially in the case where a limited amount of paired training data is available.

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

Received 20 February 2007; revised 12 March 2009; accepted 5 June 2009

