# OpenReview forum: "Hunting Blemishes: Language-guided High-fidelity Face Retouching Transformer with Limited Paired Data"
_acmmm.org/ACMMM/2024/Conference — MM2024 Poster_

### Official Review · Reviewer_EU1r · 2024-05-10

**Rating:** 4
**Confidence:** 4

**Summary:**

The paper introduces LangBRT, a novel face retouching transformer that achieves high-fidelity blemish removal with limited paired training data. It leverages vision-language pretraining for precise facial blemish detection and employs a text-prompted detection module to guide the editing process. LangBRT uses a target-aware cross-attention mechanism to edit blemish-like regions accurately while maintaining normal skin areas unchanged. Extensive experiments demonstrate its superior performance over existing methods, especially under conditions with scarce training data. The model is capable of handling diverse blemishes and provides user-defined retouching options.

**Strengths:**

1. The method proposed in this paper is very effective and has achieved satisfactory results on mainstream face retouching datasets.
2. The writing of the paper is smooth, the drawing is exquisite, and it can meet the requirements of publication.
3. The idea of using alpha clip to assist retouching is very natural and effective, which is a successful attempt.
4. The experimental results are solid with detailed analysis.

**Limitations:**

1. Authors should try to conduct experiments on different ethnic groups and try different degrees of flawed data
2. The proposed method contains too many elements. Personally, I prefer the distillation of simple and effective methods
3. Please compare recent methods dedicated to retouching, such as：
[G. Hong et.al, "HQRetouch: Learning Professional Face Retouching Via Masked Feature Fusion and Semantic-Aware Modulation,"  2023 IEEE International Conference on Image Processing (ICIP)]

**Suitability:**

3

---

### Official Review · Reviewer_fEHp · 2024-05-24

**Rating:** 5
**Confidence:** 3

**Summary:**

This paper proposes a model named LangBRT, which provides an efficient and accurate solution for facial blemish removal by combining a text-prompted blemish detection module and a target-aware cross attention mechanism. It also ensures high-quality output of the synthesized image through regularization techniques.

**Strengths:**

The content of this paper is clear, readable, the argument of the paper is comprehensive, the experiment is sufficient, and the credibility is high. The math part of the paper is logically clear.
The Text-prompted Blemish Detection is an innovative method introduced in the paper. For blemishes in specific areas such as dark circles, acne, and wrinkles, the TBD can utilize a pre-trained visual language model to obtain text prompts. It then uses an image encoder to extract features from the source image and generates corresponding mappings that indicate the blemishes related to the description.
Additionally, the paper introduces a Target-Specific Cross-Attention mechanism (TCA). The TCA is capable of injecting defect information into the transformer blocks in T, which limits the main feature transformation of the defect-like areas and guides the transformer blocks to progressively restore the clean skin in that area.
The experimental results of the paper also demonstrate that the proposed method can produce more satisfactory results in removing different types of blemishes.

**Limitations:**

1. The model proposed in the paper may have a relatively complex structure, which could cause higher computational costs during both model training and inference.
2. Figure 2 shows the description of the Discriminator but does not show the position and relationship between it and other modules in the overall framework. And there is no reference to Discriminators in the annotations.
3. The format of formula 11 looks a little strange.

Furthermore, I find that the mapping layer in the TBD of figure 2 is an ascending dimension process according to the diagram, but the size of its input and output are the same in figure 2.

**Suitability:**

3

---

### Official Review · Reviewer_y1MZ · 2024-05-24

**Rating:** 3
**Confidence:** 2

**Summary:**

This paper tackles the challenge of automatically retouching face images while reducing the dependency on paired training data. Motivated by the limitations of existing methods in handling complex and unusual blemishes, the authors propose the Language-guided Blemish Removal Transformer (LangBRT). LangBRT leverages vision-language pre-training and incorporates a text-prompted blemish detection module to identify and edit specific blemishes, while maintaining normal skin regions. The framework integrates a target-aware cross attention mechanism within transformer blocks to ensure precise blemish removal. Extensive experiments demonstrate that LangBRT outperforms state-of-the-art auto-retouching methods, showing superior performance in blemish detection accuracy and synthesis quality, even with limited training data.

**Strengths:**

1. Innovative Approach with Language Guidance: The LangBRT framework introduces a unique method that leverages vision-language pre-training for facial blemish removal. This approach reduces the dependency on paired training data and allows the model to handle specific blemishes based on textual prompts, which is a significant improvement over traditional methods that require extensive paired datasets.

2. Target-Aware Cross Attention Mechanism: The incorporation of a target-aware cross attention mechanism ensures precise editing of blemish-like regions while maintaining the integrity of normal skin areas. This mechanism enhances the accuracy and quality of the retouched images, which is critical for applications requiring high fidelity and realism.

3. Regularization for Semantic Consistency: By adopting a regularization approach that promotes semantic consistency between the synthesized image and the text description, LangBRT ensures that the final output aligns closely with the desired retouching outcome. This feature not only improves user satisfaction but also sets a new standard for the interpretability of automated retouching results.

**Limitations:**

1. Complex Implementation and Training: The LangBRT framework's reliance on advanced components like vision-language models, cross attention mechanisms, and regularization techniques makes it complex to implement and train. This complexity can be a barrier for widespread adoption, especially for users with limited computational resources or expertise.

2. The modification of ViT in this downstream task lack of the overall innovation of the related community.

3. Also, the main figure of the proposed method seems a little bit fuzzy and can be state clearer with sperate ones.

3. Limited Real-World Validation: While the paper demonstrates impressive results in controlled experiments, there is a need for more extensive real-world validation. The model's performance in diverse and uncontrolled environments needs to be thoroughly tested to confirm its robustness and adaptability.

4. Potential Scalability Issues: The framework's reliance on textual prompts and extensive feature extraction processes may pose scalability challenges when applied to large-scale datasets or in real-time applications. Ensuring efficient computation and maintaining performance at scale remains a significant challenge.

**Suitability:**

3

---

### Official Review · Reviewer_moiZ · 2024-05-25

**Rating:** 4
**Confidence:** 3

**Summary:**

The paper presents a novel approach to facial blemish removal using a Language-guided Blemish Removal Transformer (LangBRT). The proposed method effectively integrates vision-language pre-training and text-prompted blemish detection to achieve high-fidelity face retouching with limited paired data. The experimental results demonstrate the superior performance of LangBRT compared to state-of-the-art methods. Overall, the paper is well-structured and the methodology is sound. However, several areas require improvement and clarification to enhance the readability and impact of the work.

**Strengths:**

1) The novelty of the LangBRT approach is well-argued, particularly the method of combining text prompts with facial retouching.

2) The paper demonstrates high scores on the FFHQR dataset, which is a notable achievement.

**Limitations:**

1) The paper lacks a quantitative comparison with RetouchFormer: Semi-supervised High-Quality Face Retouching Transformer with Prior-Based Selective Self-Attention (AAAI 2024), which is the SOTA on FFHQR. Including this comparison would strengthen the evaluation of the proposed method.

2) The explanation of the Text-prompted Blemish Detection (TBD) module is somewhat abstract. Including a more detailed algorithmic description or pseudo-code would aid in understanding.

3) Clarify the process of how the dataset was randomly selected. A detailed explanation would provide more transparency regarding the experimental setup.

4) The paper should provide more information on the number of predefined prompts. Explain how predefined prompts are aligned with user input prompts during training, and how the model distinguishes between different types of blemishes in the same region, such as "forehead wrinkles" and "acne on forehead".

5) The experimental setup, including dataset details and evaluation metrics, is adequately described, but additional clarity on the selection criteria for the in-the-wild data would be beneficial.

6) The ablation study is comprehensive, but consider providing more visual examples of the intermediate outputs from the ablation variants to illustrate the contributions of each component more effectively.

**Suitability:**

3

---

### Meta-Review · Area_Chair_zWm6 · 2024-06-30

**Recommendation:** Accept (Poster)
**Confidence:** 5

**Metareview:**

The strengths of this paper includes Innovative use of language guidance for facial retouching, as well as clear, comprehensive argumentation and convincing experimental results. The limitation of this paper includes insufficient clarity on dataset selection, lack of comparison with RetouchFormer, limited real-world validation as well as potential scalability issues. In the rebuttal, the authors have addressed most of the concerns, hence all the reviewer gave positive recommendations. Therefore, the AC recommends to accept this paper.

---

### Meta-Review · Senior_Area_Chairs · 2024-07-10

**Recommendation:** Accept (Poster)
**Confidence:** 4

**Metareview:**

This paper received mixed ratings initially. After rebuttal, all the reviewers tend to accept the paper. SAC and AC agree with reviewers and recommend acceptance of the paper.